# Interplay between spherical confinement and particle shape on the self-assembly of rounded cubes

Da Wang [1], Michiel Hermes[1], Ramakrishna Kotni[1], Yaoting Wu[2], Nikos Tasios[1], Yang Liu[1,3], Bart de Nijs[1], Ernest B. van der Wee [1], Christopher B. Murray[2,4], Marjolein Dijkstra[1] & Alfons van Blaaderen[1]

Self-assembly of nanoparticles (NPs) inside drying emulsion droplets provides a general strategy for hierarchical structuring of matter at different length scales. The local orientation of neighboring crystalline NPs can be crucial to optimize for instance the optical and electronic properties of the self-assembled superstructures. By integrating experiments and computer simulations, we demonstrate that the orientational correlations of cubic NPs inside drying emulsion droplets are significantly determined by their flat faces. We analyze the rich interplay of positional and orientational order as the particle shape changes from a sharp cube to a rounded cube. Sharp cubes strongly align to form simple-cubic superstructures whereas rounded cubes assemble into icosahedral clusters with additionally strong local orientational correlations. This demonstrates that the interplay between packing, confinement and shape can be utilized to develop new materials with novel properties.

[1] Soft Condensed Matter, Debye Institute for Nanomaterials Science, Utrecht University, Princetonplein 5, 3584 CC Utrecht, The Netherlands. [2] Department of Chemistry, University of Pennsylvania, Philadelphia, PA 19104, USA. [3] Department of Earth Sciences, Utrecht University, Budapestlaan 4, 3584 CD Utrecht, The Netherlands. [4] Department of Materials Science and Engineering, University of Pennsylvania, Philadelphia, PA 19104, USA. These authors contributed equally: Da Wang, Michiel Hermes. Correspondence and requests for materials should be addressed to D.W. (email: d.wang@uu.nl) or to A.V.B. (email: a.vanblaaderen@uu.nl)

**S**tructuring matter by self-assembly (SA) with either nanoparticles (NPs) or micron-sized colloids has progressed significantly over the last decades[1–4]. This advance has been driven, from a materials perspective, by both improvements in the synthesis of individual building blocks as well as the realization that new collective properties can emerge in finely tuned 3D structures[3–5]. To functionalize the 3D structures, it is often desirable to structure the particles over multiple length scales. Control over the length scale of the SA as well as the crystallographic orientation is important for instance for the electronic (e.g., conductivity) and optical (e.g., polarization of light emission) properties of such materials[6–11].

One promising route to structure NPs over multiple length scales is to let the NPs self-assemble in spherical confinement, i.e., drying emulsion droplets[3]. The advantage of this, instead of methods based on the introduction of attractions through, e.g., solvophobic effects[12], is that it does not depend on the detailed surface chemistry of the particles and can thus be applied to a wider range of systems and particles with more complex shapes[13]. This has been applied to prepare so-called supraparticles (SPs) with a wide range of morphologies and length scales[5,13–20]. Lacava et al.[20] attributed the formation of icosahedral clusters of gold NPs to energetic interactions, similarly as was, e.g., found for atomic systems interacting through a Lennard–Jones potential. Later, we discovered that when less than about a hundred thousand spherical particles are confined in a drying emulsion droplet, entropy and spherical confinement alone are sufficient for the formation of a stable icosahedral cluster[14]. The resulting SPs, particles made from particles, allow for further control over for instance optical properties, e.g., SPs have a more symmetric photonic signature than a bulk crystal[17], but spherical particles adopt random orientations, and thus offer no control over the material properties that depend on orientations.

Non-spherical particles with size and shape uniformity can form a large diversity of hierarchical structures[21–26]. The SA of cubes has been investigated extensively both theoretically[11,21,22,24,27–32] and experimentally[9,11,23,24,26,33–50]. Self-assembled nanocubes can often possess distinctive properties that are quite different from those of spheres. For example, quantum dot cubes have been demonstrated experimentally and

theoretically to form NP solids with higher directional charge transport compared to spheres, as the cubes align their flat faces, thereby creating stronger coupling between neighboring NPs[51]. The tunability of the shape of crystalline cubes[23,52,53] makes such particles an ideal model system to investigate the effect of asphericity and of the flat faces on the self-assembled structures and thus further modulate their collective properties.

In this work, by integrating experiments and computer simulations, we probe how and to what extent the SA of particles in spherical confinement is affected by particle shape, by investigating shapes that interpolate between perfect sharp cubes and perfect spheres via rounded cubes. We demonstrate experimentally that in slowly drying emulsion droplets[14] sharp cubic and rounded cubic NPs self-assemble into spherical SPs. We find that locally the partially flat faces orient the nanocubes with respect to their neighbors inside the self-assembled SPs. Sharp cubes have strong orientational correlations which span the whole SP, while rounded cubes line up with each other with a short-ranged orientational order. We illustrate this concept for sharp cubic and rounded cubic NPs, but we believe our findings are applicable to a broad range of other shape-engineered single crystalline NPs as well.

## Results

**Shape of the building blocks.** We approximate the experimental particle shape by the Minkowski sum of a cube (with height $A$) and a sphere (with radius $R$), resulting in a rounded cube with height $D = A + 2R$ (Fig. 1a). The particle shape interpolates smoothly from perfect sharp cubes with an asphericity $\alpha = A/D = 1$ to perfect spheres for $\alpha = 0$. This shape is expected to more accurately describe experimentally achievable particle shapes than the often simulated superballs[27]. Particles with this shape can experimentally be achieved for a wide range of systems[16,26,34,40,44]. We synthesized sharp $Fe_3O_4$ nanocubes with a core side length of 22.7 nm[52] (24.1 nm total side length due to interdigitating ligands; Fig. 1b and Supplementary Methods Sections 2.1 and 2.3) and 9.0 nm rounded $Fe_xO/CoFe_2O_4$ nanocubes[53] (10.4 nm total side length due to interdigitating ligands; Fig. 1c and Supplementary Methods Sections 2.2 and 2.3). The shape of the hard core of the NP is well described by our

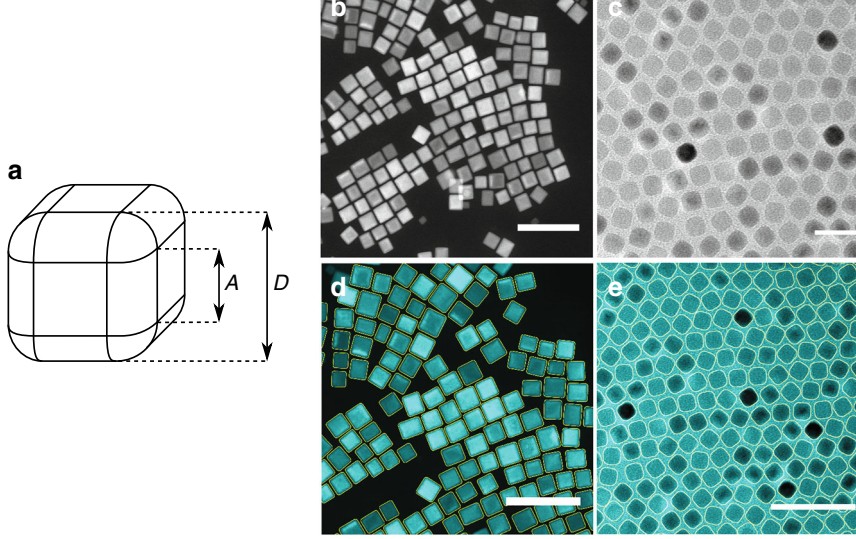

**Fig. 1** The shape of building blocks. **a** A schematic illustration of the NP shape, **b** HAADF-STEM image of sharp $Fe_3O_4$ nanocubes, and **c** TEM image of rounded $Fe_xO/CoFe_2O_4$ nanocubes, **d** EM image of nanocube cores (cyan) with the fitted shape (yellow) for the sharp $Fe_3O_4$ nanocubes and **e** same as in **d** but for the rounded $Fe_xO/CoFe_2O_4$ nanocubes. Scale bars, **b** 100 nm, **c** 20 nm, **d** 100 nm, **e** 50 nm

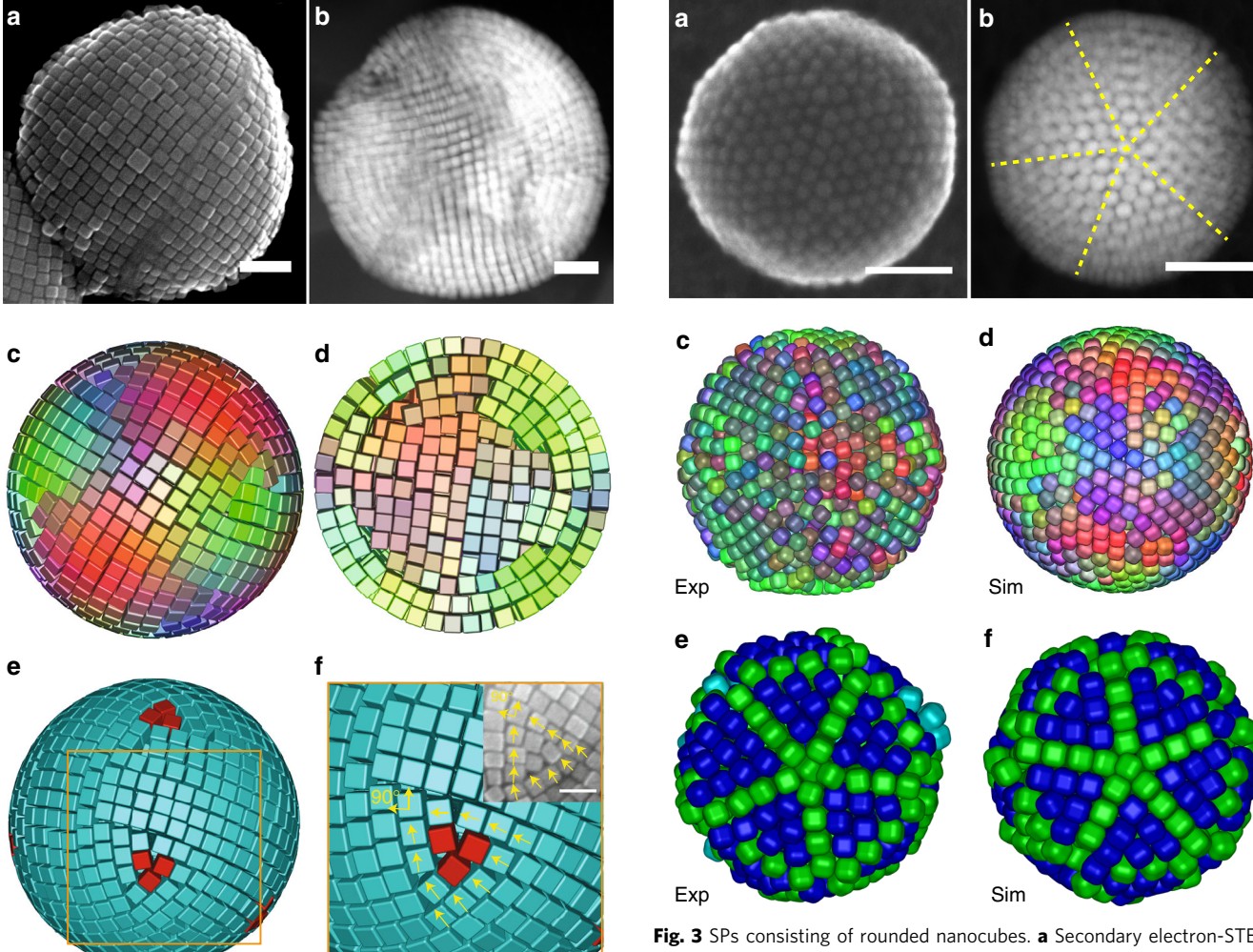

**Fig. 2** SPs consisting of sharp nanocubes. **a** SEM and **b** cryo-HAADF-STEM images of a SP containing sharp $Fe_3O_4$ nanocubes, simulation results of **c** surface, and **d** core termination of a SP resulting from the SA of sharp cubes ($\alpha = 0.8$) in spherical confinement, note that different colors represent different orientations (Supplementary Methods Sections 3 and 4). **e**, **f** topological defects as denoted by red. Inset, disclination found in experiment. For an interactive 3D view of **c** and **e**, see Supplementary Data 1 and 2, respectively. Scale bars, **a**, **b** 100 nm, inset in **f**, 50 nm

**Fig. 3** SPs consisting of rounded nanocubes. **a** Secondary electron-STEM (SE-STEM) image of a SP containing rounded $Fe_xO/CoFe_2O_4$ nanocubes, **b** 2D HAADF-STEM image of a SP of $Fe_xO/CoFe_2O_4$ nanocubes for tomography study, where the fivefold symmetry of a Mackay icosahedron, is readily visible. **c**, **e** experimental and **d**, **f** simulation results ($\alpha = 0.3$) of **c**, **d** surface and **e**, **f** core termination of a SP resulting from the SA of spherical confined rounded $Fe_xO/CoFe_2O_4$ nanocubes, which exhibits Mackay icosahedral symmetry. Note that different colors in **c** and **d** represent different orientations of the rounded cubes at the surface. For an interactive 3D view of **c**–**e**, see Supplementary Data 3–5, respectively. Scale bars, 100 nm

theoretical model: $\alpha_{core} \approx 0.73$ (Fig. 1d) for the $Fe_3O_4$ nanocubes and $\alpha_{core} \approx 0.35$ (Fig. 1e) for the $Fe_xO/CoFe_2O_4$ nanocubes.

The nanocubes are stabilized by ligands, oleic acid molecules in our case. To take into account the effect of the ligands we assume that they form a shell of constant thickness $L$ around the hard core of the particle. We have obtained the thickness of the ligand layer $L = 1.4$ nm from the particle separation by comparing the simulated radial distribution function $g(r)$ as a function of the particle separation $r$ with the experimentally obtained $g(r)$ (see Supplementary Methods Section 2.3). Taking into account the ligands, the total shape parameter $\alpha$ of the $Fe_3O_4$ nanocubes and $Fe_xO/CoFe_2O_4$ nanocubes is 0.68 and 0.30, respectively (see Supplementary Methods Section 2.3 for more details). It is likely that the NP interactions are not exclusively hard and that non-spherical NPs at the droplet interface induce more complex interactions[12,42]. However, the good correspondence between our experiments and the simulations, in which these interactions are not taken into account, indicates that the experimental SA is largely determined by entropy.

**SA of sharp cubes in spherical confinement**. We first focus on our two experimental data sets of sharp nanocubes ($\alpha \approx 0.7$; Fig. 2) and rounded nanocubes ($\alpha \approx 0.3$; Fig. 3) and compare them with computer simulations, later we will discuss the whole range of $\alpha$ values available in our computer simulations (Fig. 4 and Supplementary Methods Section 3). To investigate the self-assembled superstructure of sharp $Fe_3O_4$ nanocubes in drying emulsion droplets, we show scanning electron microscopy (SEM) and cryo-high angle annular dark field scanning transmission electron microscopy (cryo-HAADF-STEM) images of self-assembled SPs containing sharp $Fe_3O_4$ nanocubes (Fig. 2a, b and Supplementary Methods Section 2.4). At the outside of the SP, the nanocubes are all aligned with their flat faces (i.e., {100} facets) towards the confining interface. In the inner core, the nanocubes form a simple-cubic (SC) lattice very similar to the bulk crystal of perfect cubes[27]. To identify the physical mechanism that drives the assembly, we performed Monte Carlo (MC) simulations of hard cubes with a range of $\alpha$ values in spherical confinement, and slowly shrunk the confining sphere,

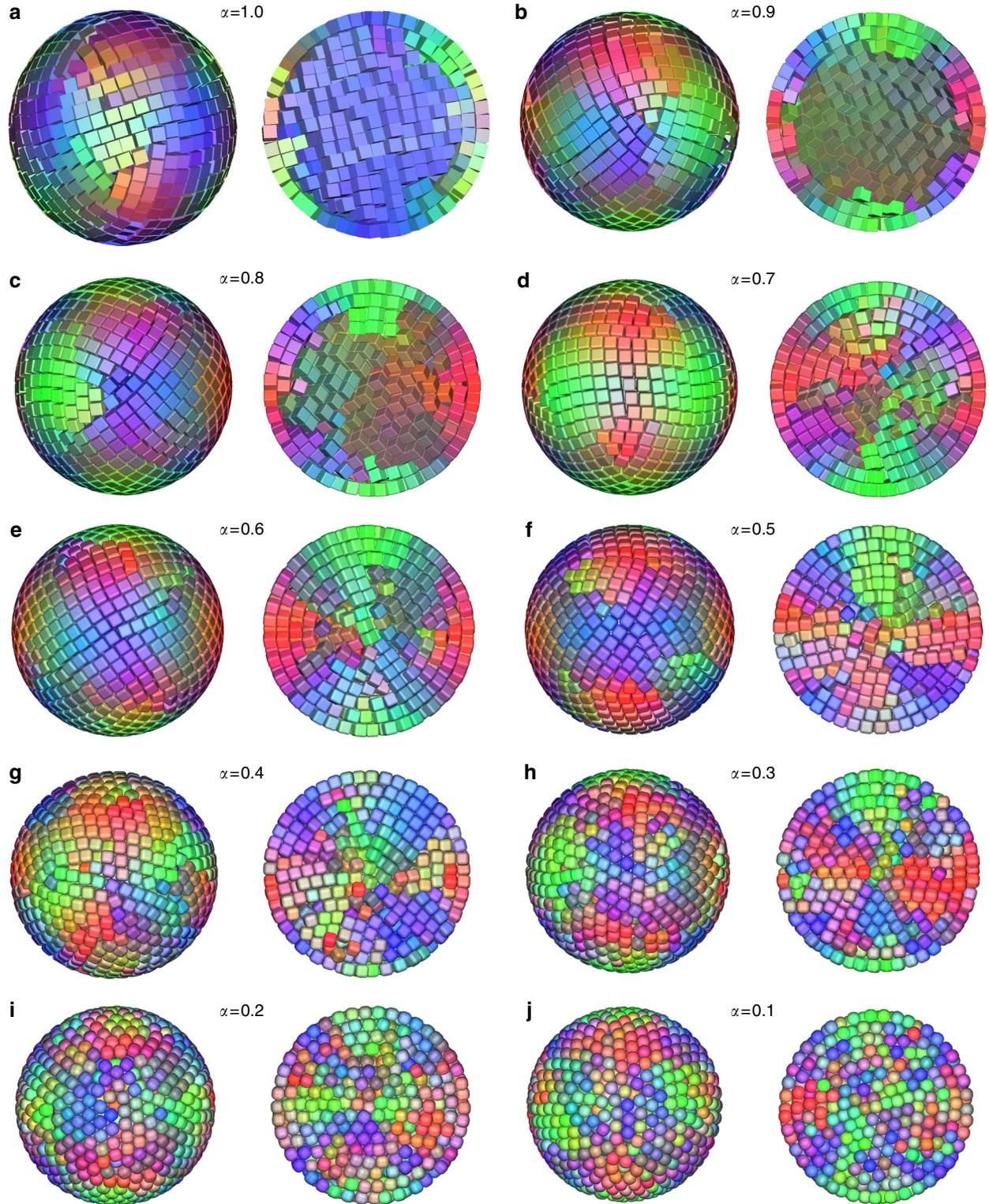

**Fig. 4** Clusters from computer simulations with varying asphericity or roundness $\alpha$. Fully compressed configurations as obtained from computer simulations with 2000 particles in spherical confinement with different shape parameters. **a–j**: $\alpha$ = 1.0 (perfect cube), 0.9, 0.8, 0.7, 0.6, 0.5, 0.4, 0.3, 0.2, and 0.1, respectively. Different colors represent different orientations. For interactive 3D views, see Supplementary Data 1 and 4–9

thereby mimicking the evaporation process (Fig. 4 and Supplementary Methods Section 3). For $\alpha > 0.5$, the cubes on the outside align with the interface, and form a distorted square lattice (Figs. 2c and 4a–e). For $\alpha > 0.75$, the cubes in the core form a (twisted) SC lattice which connects smoothly to the outer layer at the six {100} facets (Figs. 2d and 4a–c). At the points where the

eight {111} facets of the central cubic lattice meet the surface aligned layers defects are formed. Further investigation of our simulation results reveal that eight topological defects with winding number of $\pi/2$ are located at these points, which is in perfect agreement with our experimental findings (Fig. 2e, f). Li et al.[54] have studied the 2D assembly of squares on a spherical

surface. The defect positions observed by them differ from the positions we observe for sharp cubes. For sharp cubic particles, we observe 8 defects in a cubic arrangement (positioned at the vertices of a cube) while Li et al. observe defects in an square antiprism arrangement (positioned at the vertices of a square antiprism or a twisted cube). In 2D SA on a curved surface, the position of the defects is solely determined by the interactions between the defects through this 2D layer. As shown by Li et al. in this case, the repulsion between the defects will cause the defects to position themselves as far away from each other as possible and to form a square antiprism arrangement. In our case, the defects do not only interact through the surface but also through the particles layers below the surface layers which includes the cubic lattice in the bulk of the SP. It is this interaction with this cubic lattice that favors the positioning of the topological defects near the vertices of this cubic lattice and causes the cubic arrangement of defects. This is strengthened by the fact that we always observe the surface defects near the corners of the central cubic lattice. Furthermore this cubic arrangement of the defects transitions to a square antiprism arrangement of defects when the SP looses its SC core when the cubes become more rounded. The fact that we see a SC core in our experiments with $\alpha = 0.7$ while we only observe this in simulations for $\alpha > 0.75$ indicates that there are small differences between the simulations and the experiments, most likely resulting from the polydispersity of the nanocubes and/or possibly a local difference in the ligand density, e.g., on the corners of the nanocubes (Supplementary Methods Section 2.3). Our simulations demonstrate that a (twisted) SC core and curved shells arise spontaneously for hard sharp cubes in spherical confinement, and reveal striking agreement with the experimental observations.

**SA of rounded cubes in spherical confinement**. To investigate the role of the NP shape, we focus on the SA of rounded $Fe_xO$/$CoFe_2O_4$ nanocubes ($\alpha \approx 0.3$) in drying emulsion droplets (Supplementary Methods Section 2.5). Figure 3a and b shows the structure of SPs formed of these rounded nanocubes. To analyze the structure of these SPs, we extracted both the positions and orientations of the nanocubes from 3D electron microscopy (EM) tomography (see Fig. 3c, e and Supplementary Movies 1 and 2), using an advanced particle tracking technique, for which we developed new fitting routines. The fitting exploits the symmetry of the NPs, which is sufficiently general to be extended to other particle shapes and provides a significant improvement to the codes recently developed to determine the positions and orientations of spherical[55] and rod-like particles[56]. Near the surface of the SPs, the nanocubes form defect-rich hexagonal layers (Fig. 3c). Figure 3e shows the core of the SP, with different colors indicating face-centered-cubic (FCC) (dark blue), hexagonal close-packed (HCP) (green), and fluid (cyan) stacked particles as identified by a bond-orientation order parameter[57,58]. The experimental SP is not perfectly spherical but is slightly faceted which can be attributed to the deformable cyclohexane–water interface. The characteristic fivefold symmetry of a Mackay icosahedron is clearly visible in the core, which consists of twenty deformed FCC ordered tetrahedral domains (Fig. 3e)[14,55]. The adjacent {111} facets of the tetrahedral domains form twinning HCP planes. Near the surface, there are a few layers that are more disordered, similar to SPs of spheres[14].

To explore whether the structural formation is entropy-dominated, we performed MC simulations of hard rounded cubes ($\alpha = 0.3$) in a slowly shrinking spherical confinement. Figure 3d shows the final (arrested) state. The cubes on the outermost layer of the SP align with their flat faces parallel to the smooth surface. In Fig. 3f, the outer layers are removed and the

particles are colored according to their stacking: green and dark blue represent HCP and FCC, respectively. This reveals the fivefold symmetry of the icosahedron in the center of the SP, which shows a striking similarity with the experimentally observed structure (Fig. 3e).

Recently, Pb-chalcogenide (PbSe, PbTe, and PbS) systems have been assembled into superlattices by oriented attachment via specific facets in 2D (or semi-2D)[7,10,42]. To investigate if the local orientation can be controlled in 3D inside SPs by other semiconductor NP systems, we studied the SA of rounded 10.7 nm semiconductor PbSe nanocubes[59] ($\alpha \approx 0.3$; 12.1 nm total side length due to interdigitating ligands; Supplementary Methods Section 2.2) in a drying emulsion droplet. Consistently, all rounded PbSe nanocubes assembled into crystalline SPs with fivefold icosahedral symmetry (Supplementary Methods Section 2.5), similar to what we found in the $Fe_xO$/$CoFe_2O_4$ system. In addition, the emergence of the icosahedral symmetry in this rounded PbSe cube system is in accordance with our assumption that the magnetic interaction is not an important parameter for the formation of the icosahedral SPs.

**Orientational correlations of cubes with varying roundness in spherical confinement**. To gain insight into the effect of the cube roundness and its influence on the orientational correlations of cubic NPs in spherical confinement, we plot the cubes with colors determined by their orientations (Fig. 4 and Supplementary Methods Sections 3 and 4). To quantify these visual observations, we also plot the orientational correlation function $g_{or}(r)$ as a function of the distance $r$ between the cubes in Fig. 5. For perfectly aligned cubes the orientational correlation $g_{or}(r) = 1$, for randomly oriented cubes $g_{or}(r) = 0$ and negative values ($g_{or}(r) < 0$) indicate anti-alignment. For sharp cubes $\alpha > 0.75$, most of the SP has a single color (e.g., Fig. 4a–c) and $g_{or}(r) > 0$ indicates a strong orientational correlation across the whole SP (Fig. 5). For $\alpha < 0.75$, there are patches of a single color visible (e.g., Fig. 4d–j) and the peak in the $g_{or}(r)$ at $r/D = 1$ demonstrates that the rounded cubes still have a preference to align with their neighbors (Fig. 5). At larger distances the color changes quickly and the orientational correlation decays to nearly zero at an average of $4D$ of the cubes. The transition between these two regimes (at $\alpha = 0.75$) corresponds to the point where the SP loses the SC core (Fig. 6). The rapid decay of the orientational correlation is most likely due to the decay in bond order. The very weak anti-alignment observed in the computer simulations for $\alpha < 0.8$ is

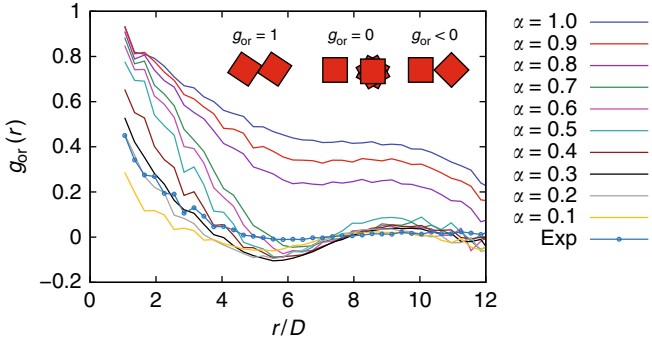

**Fig. 5** Orientational correlations of cubes in spherical confinement. Orientational correlation functions $g_{or}(r)$, for ten different shape parameters $\alpha$ as a function of the radial distance $r$. Inset, an illustration of alignments of two cubes at contact. Here $r$ is the radial distance and $D$ is the side length of the cube which is set to unity. Note that the curve labeled as exp denotes the $g_{or}(r)$ of the experiment on the rounded $Fe_xO$/$CoFe_2O_4$ nanocubes ($\alpha = 0.3$)

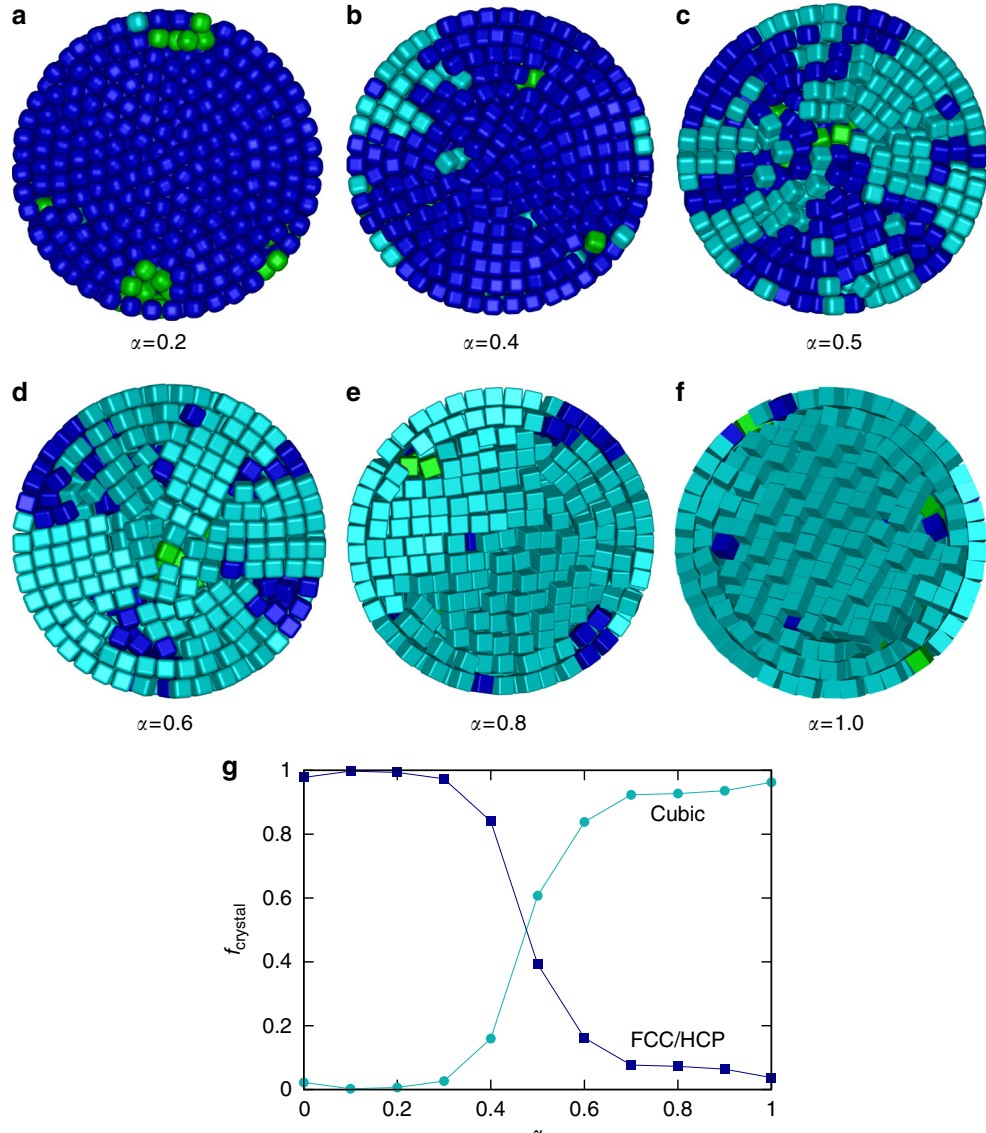

**Fig. 6** Transition from FCC/HCP to SC. Cut-through views of **a–f** typical configurations of SPs of cubes with different asphericity $\alpha$ = 0.2, 0.4, 0.5, 0.6, 0.8, and 1.0 as obtained from simulations. The dark blue cubes have FCC/HCP order and the cyan particles have cubic order. The green cubes are not crystalline. **g** The fraction of crystalline cubes that is in an environment with cubic symmetry (cyan) and the fraction of crystalline cubes that is in an environment with FCC/HCP symmetry (dark blue) as a function of $\alpha$

absent in the experiments. We expect that this is caused by the slight deformation (flattening of the {111} facets) of the experimental SP. For $0.75 > \alpha > 0.05$ the local orientational order of the particles is still strongly correlated with the local crystalline order, although the structure of the SP is very similar to the structure formed by spheres (e.g., Fig. 4j). When we look at the SA (in simulations) of these cubes, we see that the cubes can still rotate freely when the crystal lattice is formed and only later when the density is increased further the particles stop rotating, indicating a transition from a plastic crystal, without orientational order, to a crystal with orientational order. This is what is expected from the phase diagram[27], however, there are distinct differences between the so-called C0 phase (deformed FCC crystal) expected in bulk and the structure observed in confinement, indicating that the icosahedral defect structure is not compatible with the transition from FCC to C0.

**SP structural transition from FCC/HCP to SC upon changing cube roundness.** To understand the structural transition between

the rounded and the sharp cubes, we performed computer simulations at varying shape parameters $\alpha$. In Fig. 6a–f, we show cross-sections of the SPs formed by 2,000 cubes with increasing asphericity $\alpha$. We determined for all cubes if they are crystalline (cyan and dark blue) or not (green). For the crystalline cubes, we characterized the symmetry of their neighbors to determine whether the cubes are in a cubic (cyan) or in a FCC/HCP (dark blue) environment. In Fig. 6g, we plot the fraction of crystalline cubes in each environment as a function of $\alpha$. For $\alpha < 0.4$, nearly all cubes are FCC or HCP stacked, although there are some less ordered fluid-like cubes near the twinning planes (Fig. 6a). For $\alpha = 0.4$, the cubes in the outer two layers start to display cubic order (Fig. 6b). For $\alpha = 0.5$, the structure consists of wedges of either cubic or FCC/HCP stacked cubes (Fig. 6c). For $\alpha = 0.6$, most cubes exhibit cubic symmetry and the structure consists of multiple wedge-shaped domains with cubic symmetry (Fig. 6d). For $\alpha \geq 0.7$, we observe a SC core and a series of spherical shells surrounding it (Fig. 6e, f). Amazingly, small changes in cube shape already alter the self-assembled superstructures.

## Discussion

To conclude, we have shown that the flat faces of the cubic NPs induce orientational order in self-assembled SPs. This might lead to novel crystal structures by inducing directed attachment between neighboring particles and can potentially be used to tune the near-field coupling inside self-assembled superstructures. In addition, we demonstrated by experiments and computer simulations that the SA of cubes in a spherical confinement displays a structural transition from defected SC clusters to clusters with icosahedral symmetry upon changing the NP shape from sharper cubes to more rounded cubes. Quantitative EM tomography on the SPs was essential to determine not only the structure of the SPs on the single NP level, but also to analyze the orientational correlations of the rounded cubes and to compare these with computer simulations. There are two organizing principles that we expect to hold more generally for hard convex-shaped particles. The first principle is that flat faces tend to align particles, both with respect to each other as well as with other flat surfaces. This alignment is strong as it limits 2 out of the 3 rotational degrees of freedom and has an important role at high volume fractions. The second principle is that sharp corners have an important role in keeping neighboring particles at a distance, and even a small amount of rounding of these corners can drastically change the phase behavior. Furthermore, in confinement there is a competition between the bulk and the surface-induced structures, both orientationally as well as positionally. Due to this competition equilibrium defects emerge that can have a complex topology whose details will depend on the symmetries of the equilibrium crystal structure and the free-energy cost of the different defect types. Therefore, we expect that our findings on sharp cubic and rounded cubic NPs to not only be of interest to the SA of micron-sized cubic-shaped particles, but also to lead to new opportunities for a broad range of other shaped crystalline faceted NPs. As such, manipulating particle shape to control local orientational order would help one to construct complex hierarchical structures with strong couplings, thus enabling additional control over the collective properties of such new materials.

## Methods

**Syntheses**. Overall, 22.7 nm sharp $Fe_3O_4$ nanocubes[52], 9.0 nm rounded $Fe_xO$/$CoFe_2O_4$[53] and 10.7 nm rounded PbSe nanocubes[59] were synthesized with minor modifications according to literature methods. As-synthesized NCs were purified by isopropanol and were dispersed in cyclohexane with desired weight concentrations. All NCs used in our current work are washed well before SA according to their original experimental protocols. Thus one can assume the amount of free ligands to be negligible after proper washing so that they do not play a significant role during the SA. Detailed information regarding to syntheses can be found in Supplementary Methods Sections 2.1 and 2.2. High-resolution TEM (HRTEM) images and additional EM micrographs of synthesized NCs can be found in Supplementary Figs. 1–3.

**Experimental SA of sharp and rounded nanocubes in spherical confinement**. For a typical sharp nanocube SA in a confinement experiment, 8.0 mg of sharp $Fe_3O_4$ nanocubes were dispersed in 1.0 mL of cyclohexane and added to a mixture of 400 mg of dextran and 23.6 mg of sodium dodecyl sulfate (SDS) in 10 mL of deionized water (DI $H_2O$). The resulting emulsion was agitated by shear with a shear rate of $1.04 \times 10^5 \, s^{-1}$, using a Couette rotor-stator device (gap spacing 0.100 mm) following the procedure and home-built equipment described by Mason and Bibette[60]. The emulsion was then evaporated at room temperature (RT) using a VWR VV3 vortex mixer for 48 h. The resulting SP suspension was purified by centrifugation with a speed of 2500 rpm for 15 min using an Eppendorf 5415C centrifuge, followed by redispersing in DI $H_2O$. The above-mentioned procedure was repeated twice. The experimental procedure of the rounded nanocubes SA is similar to that of sharp nanocubes, except that 6.5 mg of rounded nanocubes were dispersed in 1.0 mL of cyclohexane and added to a mixture of 400 mg of dextran and 70 mg of SDS in 10 mL of DI $H_2O$, and the resulting emulsion was agitated by shear with a shear rate of $1.56 \times 10^5 \, s^{-1}$. The size of the self-assembled SPs ranges from ~100 to ~800 nm. Experimental details are available in Supplementary Methods Sections 2.4 and 2.5. Additional EM micrographs of self-assembled SPs consisting of rounded nanocubes can be found in Supplementary Figs. 6 and 8.

**EM sample preparation and measurements**. Conventional TEM imaging was performed on a FEI Tecnai 12 with a tungsten tip, operating at 120 kV. For STEM a FEI Tecnai 20F with a Field Emission Gun (FEG) was used, operating at 200 kV in SE-STEM mode. HRTEM, 2D and 3D HAADF-STEM measurements were obtained with a FEI Talos F200X TEM, equipped with a high-brightness field emission gun (X-FEG) and operated at 200 kV. SEM imaging was performed on a FEI Helios NanoLab G3 UC focused ion beam (FIB)-SEM in SE mode at 15 kV.

Cryo-TEM and cryo-HAADF-STEM measurements were performed using a FEI Tecnai 20F with a FEG at 200 kV. Liquid samples (3 μL) were dropped on a Quantifoil (2/2, 200 mesh) copper grid. The thin film specimen was instantly frozen in liquid ethane using a Vitrobot Mark2 plunge freezer and cooled to ~90 K. After ethane was removed, the frozen sample was inserted into a Gatan cryo-transfer holder. The imaging was carried out at temperatures around 90 K in TEM (see Supplementary Fig. 4) and HAADF-STEM mode. Cryo-SEM imaging was performed on a FEI Helios NanoLab G3 UC FIB-SEM in SE mode, operating at 15 kV under cryogenic condition. Liquid samples (3 μL) were dropped on a copper substrate, followed by blotting with filter paper to remove the bulk of the liquid. The thin film specimen was frozen in liquid nitrogen and cooled to approximately 77 K. Next, the sample was transferred into the microscope and imaging was carried out at temperatures around 90 K (see Supplementary Fig. 5b).

To prepare a sample for EM tomography analysis, 3 μL of the SPs suspension in DI $H_2O$ were deposited on a Quantifoil (2/2, 200 mesh) copper grid and plunge frozen in liquid ethane using a Vitrobot Mark2 plunge freezer at temperatures around 90 K. The sample was then freeze-dried over a period of 8 h under vacuum at 177 K and subsequently allowed to warm to RT prior to electron microscopy analysis. For the SPs consisting of rounded $Fe_xO$/$CoFe_2O_4$ nanocubes, a Fischione model 2020 single tilt holder was used for the acquisition of the tilt series within a tilt range from −70° to +70°, and with an increment of 2° (see Supplementary Movie 1).

The tilt series were aligned using cross-correlation routines implemented in Fiji v1.51p (http://fiji.sc/) and TomoJ[61]. The reconstruction was performed using the Simultaneous Iterative Reconstruction Technique (SIRT)[62] algorithm in TomoJ 2.31 (see Supplementary Movie 1). Segmentation of the tomograms was carried out mainly through thresholding and marker-based watershed transformation in Avizo 9 (Thermo Fisher Scientific-Electron Microscopy Solutions, https://www.fei.com/software/avizo-for-materials-science/), as well as 3D volume rendering of the reconstructed SPs (see Supplementary Fig. 7 and Supplementary Movie 2).

**Simulations**. We performed Monte Carlo (MC) simulations of hard rounded cubes with height $D$ and rounding parameter $\alpha$ in the canonical (NVT) and isothermal-isobaric (NPT) ensembles. To model the spherical confinement we used an impenetrable hard spherical wall. To mimic the evaporation of the solvent from the droplets the diameter of the spherical confinement was slowly reduced in time. We used bond order parameter $Q_4$ to identify the local symmetry of the particles. More simulation details as well as the methods used to color the particles with respect to their orientation are reported in Supplementary Methods Sections 3 and 4 and Supplementary Figs. 9–11.

**Analysis of defects**. It is not possible to tile the surface of a sphere with a square grid of squares without introducing topological defects. The cubes in our simulations and experiments were nearly always found to align with a square face towards the spherical shell and will thus also form topological defects. The topological defects are characterized by their winding number[54], which is 90° or $\pi/2$ in this case (the rotational symmetry of a square), see Fig. 2e, f. To obtain the curvature of a sphere a total topological charge of $4\pi$ or at least eight topological defects are required. To automate the analysis of the topological defects on the surface of the SPs, we analyzed the orientation of the nanocubes with respect to their neighbors. The topological defects can be identified by tracing a closed chain of particles and tracking the smallest rotation to rotate a particle to the orientation of the next particle in the chain. This is illustrated in Fig. 2f, the particles marked with yellow arrows form a chain and the yellow arrows show the orientation of the particle. If one starts tracing the orientation at the top left particle and follow the chain until one is back at this particle, the orientation has changed by 90° ($\pi/2$). This means that this chain surrounds a topological defect of charge $\pi/2$. In Fig. 2, we show one of these topological defects on the surface for $\alpha = 0.8$. The red particles in Fig. 2f of the main text are the particles for which the chain formed by their neighbors surrounds one of these topological defects. The topological defect is a point on the surface of the sphere that lies between these red particles.

**Determining the position and orientation of the nanocubes**. The locations and orientations of the individual NPs were determined from the 3D reconstructed tomograms (Supplementary Fig. 12). As a first step we apply a 3D Gaussian blur (with a radius of 1.6 pixels) to the reconstructed data to get rid of the noise. The number of particles $N$ and their rough positions $\tilde{r}_i$ were found by a 3D centroiding algorithm similar to the one used by Crocker and Grier[63]. The pixels in the picture are then divided into regions $R_i$ each belonging to a single particle by use of a watershed algorithm. Region $R_i$ is a set of pixels $R_i = \{\{I_1, x_1\}, \{I_2, x_2\}, \ldots \{I_m, x_m\}\}$ with intensity $I_n$ and position $x_n$ for which water flowing along the steepest ascent would flow to a single local intensity maximum associated with particle $i$. We do

not take pixels with an intensity below a threshold of 0.05 (relative to a maximum intensity of 1) into account.

The center of mass, or actually center of intensity, is then measured for all pixels belonging to a single particle to determine the exact position of the particle

$$\mathbf{r}_i = \frac{1}{I_{tot}} \sum_{n=0}^{pixels \in R_i} I_n \mathbf{x}_n, \tag{1}$$

where $I_{tot}$ is the sum of all intensities in $R_i$. To obtain accurate coordinates this and all following steps are performed on the unfiltered data. To determine the orientation, we use the technique described in the literature[64]. We describe the intensity distribution around the center of mass in terms of spherical harmonics. As we know the symmetries of our particles, we know that we only need to look at $l = 4$ and $l = 6$. We can then align the particle by maximizing the correlation between the spherical harmonics of the found particle and the same expansion in spherical harmonics of a reference particle by rotating the expansion using Wigner D matrices. In principle, the intensity can be used directly but we found that for the reconstructed tomography images the gradients are more reliable. The details of the algorithm will be discussed in a later publication.

**Crystalline domains**. We used a bond order analysis as described by Steinhardt et al.[57] to characterize the local symmetry of nearest neighbors of the particles (see Fig. 6). To determine the neighbors, we used a bond length of $1.3D$, with $D$ the side length of the particles, as there are no 'real' bonds for hard particles. This includes the first peak of the $g(r)$ but excludes the second peak both for perfect cubes as well as for perfect spheres (Supplementary Fig. 13). To identify crystalline particles, we used the approach as described by Ten Wolde et al.[58] and calculate correlations between a particle and its neighbors by taking the dot product between the **q** vectors. Crystalline bonds are defined as neighbors with which the dot product of the $\mathbf{q}_6$ vectors is larger than 0.5. For the face-centered-cubic (FCC)/hexagonal close-packed (HCP) particles, we defined a particle as crystalline when it has 5 or more crystalline bonds. For the particles in a cubic lattice, we defined a crystalline bond when the dot product between the $\mathbf{q}_4$ vectors was larger than 0.5. We defined a particle to be crystalline when it had 3 or more crystalline bonds (there are only 6 nearest neighbors in a perfect crystal, instead of 12).

Although superballs[27] do not have exactly the same shape as the rounded cubes studied here we do expect their phase diagrams to be very similar. From the bulk, there are several different crystal structures that we expect to encounter. Perfect cubes form a SC crystal, which continuously transforms to a so-called C1 phase[27] (deformed SC crystal, see Supplementary Fig. 14a) when the cubes are rounded. Perfect spheres form an FCC crystal with a small free-energy difference with HCP[65]. When these particles develop flat faces ($\alpha > 0$) the so-called C0 phase (deformed FCC crystal, see Supplementary Fig. 14b) appears at high packing fraction. As SPs with an icosahedral symmetry can approximately be seen as being build up from FCC stacked 'pie-points' with local HCP stacking at the connections, we used the FCC/HCP discrimination also for the icosahedrally arranged SPs. To distinguish if particles are cubic stacked or FCC/HCP stacked we calculated both $Q_4 = \mathbf{q}_4 \cdot \mathbf{q}_4$, high in cubic symmetry, as well as $Q_6$, high in HCP. When $Q_4 + 0.17 > Q_6$, we defined a particle as cubic stacked and when $Q_6 > Q_4 + 0.17$, we defined a particle with a FCC/HCP stacking. To distinguish between FCC and HCP, we used the $W$ order parameter[66].

In bulk simulations of rounded cubes, we observed upon compression a transition from a plastic FCC crystal to a fully aligned C0 crystal. One of the main characteristics of the C0 phase is that the nearest neighbors do no longer align perfectly with the faces of the cubes but are shifted slightly off center. In Supplementary Fig. 15a, the position of the nearest neighbors is shown for the plastic crystal FCC phase and in Supplementary Fig. 15b for the C0 phase of rounded cubes in bulk for $\alpha = 0.3$. Even in the case of the plastic crystal, the particles already have a preference for certain orientations, they avoid pointing their corners towards any of their neighbors. In the C0 phase the 4 spots (Supplementary Fig. 15b) in the center are caused by the off center position of these nearest neighbors resulting from the deformation of the unit cell and the alignment of the particles. However, when we compressed the same rounded cubes in spherical confinement we did not observe this transition, instead the particles remained in a more symmetric FCC/HCP position. This is illustrated in Supplementary Fig. 15c (simulation) and d (experiment), where we plot the positions of the nearest neighbors for rounded cubes in confinement and observe a single bright spot in the center (neighbors aligned to the center of the faces).

**Orientational correlations**. The orientation of cube $i$ is stored as a rotation matrix $\mathbf{M}$. To quantify the orientation of the cubes with respect to neighboring cubes we want to calculate the minimum angle $\alpha_{ij}$ required to rotate a cube with orientation $\mathbf{M}_i$ into the same orientation as a cube with orientation $\mathbf{M}_j$. The angle between two orientation matrices can be obtained from the angle axis $1 + 2 \cos\alpha_{ij} = \text{Tr}\left(\mathbf{M}_i^T \mathbf{M}_j\right)$ but this would not take into account the symmetries of a cube. To do this, we need to calculate the symmetry group of the cube (or octahedron) $S_4$, which contains 24 rotations $\mathbf{M}_k^{S4}$ that preserve the orientation of a cube. To obtain the minimum angle, we then calculate the angle between all identical representations of the

orientations of the cubes

$$\alpha_{ij} = \min\left\{ f\left(\mathbf{M}_i^T \mathbf{M}_1^{S4} \mathbf{M}_j\right), f\left(\mathbf{M}_i^T \mathbf{M}_2^{S4} \mathbf{M}_j\right), \cdots \right\}, \tag{2}$$

where $f(x) = \text{acos}(\text{Tr}(x)/2 - 1/2)$ and the minimum is taken over all 24 members of the symmetry group.

**Connecting particle color with its orientation**. To obtain a visually interpretable effect, we color the particles by calculating the angle between the particle and three reference particles with different orientations. Detailed information can be found in Supplementary Methods Section 4.

**Data availability**. The authors declare that all the relevant data are available within the paper and its Supplementary Information file or from the corresponding author upon reasonable request.

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

## Acknowledgements

We thank L. Filion, F. Smallenburg, S. Dussi, X. Xie, C. Xia, and J. L. Peters for fruitful discussions. X. Xie is acknowledged for assisting with HRTEM measurements. J.D. Meeldijk is acknowledged for discussions of cryo-EM experiments. D.W., E.B.v.d.W., and A.v.B. acknowledge partial financial support from the European Research Council under the European Union's Seventh Framework Programme (FP-2007–2013)/ERC Advanced Grant Agreement 291667 HierarSACol. M.H. and R.K. were supported by the Netherlands Center for Multiscale Catalytic Energy Conversion (MCEC). Y.L. acknowledges the Sustainability project between the faculties of Science and Geosciences of Utrecht University. N.T. and M.D. acknowledge financial support from an NWO-ECHO grant. This material is based upon work supported by the National Science Foundation under Grant No. CHE-1709827. The authors acknowledge the EM square center at Utrecht University for the access to the microscopes.

## Author contributions

D.W., B.d.N., and A.v.B. initiated the experimental part of the project. D.W., R.K., B.d.N., and Y.W. carried out nanocrystal syntheses under supervision of A.v.B. and C.B.M. D.W. performed self-assembly, (Cryo) EM imaging and electron microscopy tomography studies under supervision of A.v.B. M.H. developed advanced particle tracking technique under supervision of A.v.B. N.T. initialized computer simulations under supervision of M.D. M.H. performed computer simulations and data analysis. E.B.v.d.W. contributed to initial crystal structure analysis under supervision of A.v.B. Y.L. contributed to (Cryo) SEM imaging and 3D representation of tomographic reconstruction. D.W., M.H., and A.v.B. co-wrote the paper. All authors analyzed and discussed results.
