## [Peer Review File · Nature Communications]

Reviewers' comments:

Reviewer #1 (Remarks to the Author):

In this paper, the authors presented a comprehensive investigation of the self-assembly of cube-like particles with spherical confinement using both experiments and computer simulations. This problem is of relevance to self-organization of nano-particles in drying emulsion droplets, and the resulting assembly structures determine their physical properties. Motivated by experimentally realizable particles, the authors investigated both perfect cubes as well as cubes with rounded corners, which are parameterized by the asphericity α . As the particle shape varies, the authors observed a rich spectrum of assembled superstructures. For example, at the cube limit, strong face-face orientational correlations are observed and a significant portion of the superstructure possess "cubic" symmetry as observed in bulk assembly, but with topologically necessary defects. On the other hand, for rounded cubes, the superstructure typically exhibits an icosahedral symmetry, with even richer local correlations.

The paper is of great topical interest and well written. It demonstrates the richness of the assembled superstructures via the interplay of particle shape and geometrical confinement, and provides guidance for experimental realization of various superstructures. Therefore, its publication is recommended. However, the authors may want to consider the following suggestions and comments, which are all optional and up to the authors.

Apparently, the superstructures generated using MC simulations were associated with a very slow shrinking rate for the spherical confinement. How does this shrinking rate affect the overall symmetry and local order of the packing? It can be expected that less ordered packing will be obtained for rounded cubes, but does the overall structure still possess the icosahedral symmetry, with a "glassy" sub-regions?

Did the authors investigate the effects of varying the number of particles on the overall assembly structures? In the infinite system limit, it can be imagined that the effect of the spherical confinement will be only localization to a very thin surface layer of the entire system. And within the system, the structure should be very similar, if not identical, to the bulk assembly structure. For finite system, the effects of the confinement would penetrate deeper in the system, and it would be very interesting to quantify this "penetration depth" if possible. For example, in Fig.6 the authors showed the fraction of local structures consistent with the bulk assembly. It would be also interesting to show such a fraction, but as a function of distance to the sphere confinement center, if the authors also think this would be helpful.

It would be helpful to put forth concise qualitative organizing principles that summarize the findings.

When the referee quickly looked at the figures for the first time, it seemed that the authors investigated dense packing of the particles on Riemann (spherical) surface (instead of a bulk packing with spherical confinement). How does the outer layer of the packing in the confinement differ from that on the spherical surface? Any observed differences should stem from the effects of the particles inside in the confinement, but it would be interesting to comment on this point as well.

Reviewer #2 (Remarks to the Author):

The manuscript by Wang et al. entitled "The interplay between spherical confinement and particle

shape on the self-assembly of sharp and rounded cubes" discusses the assembly of cubic inorganic nanoparticles inside evaporating emulsion droplets. The authors, well-known experts in the field, combine simulation and experiments and provide a well-written and clearly illustrated account of their work. This is an important contribution in my opinion, in particular since there is an ongoing discussion in the field on the relative magnitudes of entropic versus enthalpic contributions in nanoparticle assembly, and this contribution gives important clues towards a complete picture. There are some shortcomings, mainly when it comes to the comparison of experiment and simulation and possible implications for the understanding of the assembly process. But the quality of data is such that I believe that the manuscript should be accepted after suitable revision.

The introduction mentions several important publications on the assembly of particles in confined spaces but misses some that I find relevant in this context. I think the authors should mention the recent work of Agthe published in a Nano Letter in 2016 that followed the assembly of maghemite nanocubes in a levitated droplet in situ, and possibly discuss their results in view of the experimental (time-dependent) scattering data. It could also be useful to mention that LaCava et al argue, in a Nano Letter from 2012, that their nanoparticles assume minimum-energy arrangements that are similar to Lennard-Jones clusters (and thus not entropically dominated).

The technical details both in the main article and in the supporting information are very thorough and clear, both in terms of experimental procedures and simulation methods; the authors should be lauded for their diligence. I found the discussion of the ligand shell structure on page 3 of the SI particularly interesting, although it leads to some additional questions (see below). If the authors have TGA data on the particles or other data that could indicate the ligand shell density and the amount of free surfactant and other molecular solutes (if any) it would be useful to add it.

In the main part, the authors clearly show that rounded corners have a strong effect on particle assembly. Their discussion of this point is convincing and combines simulations and experiments in a very useful way. It would possibly have been more didactic to structure the two central figures 2 and 3 more similar - I was first confused because panels c and d look similar but are once outer/inner views and once experiment/simulation - but this is probably a matter of taste. I do find it very important, however, to clearly indicate whether the data in figure 5 is based on simulation and experiments - and ideally to compare both. Both datasets should be available, why not plot them in the same way? More in general, the discussion should say more on the similarities and differences between simulations and experiments. The authors correctly write (in the SI) that the sensitivity of assembly on particle shape (and perhaps interaction?) could be used to understand the assembly process better. A more stringent comparison in this manuscript could give first indications on the role of the ligand shell, for example, even if detailed investigations are to follow in a forthcoming contribution.

The discussion of the ligand shell is, in my opinion, the only true shortcoming of the manuscript at hand. The authors mention the shell on these particles in multiple instances and correctly point out that they somewhat smoothen the face. They only discuss the ligand shell thickness in the SI, however, and it is not clear which contributions they expect, how "round" the particles with the ligand shell really are, or at least how thick the ligand shell is (the SEMs seem to imply thin shells, but this is the dry collapsed state). Given the discussions on entropic versus enthalpic contributions in assembly, I think that a more thorough discussion would be in order. If the outcome is that the ligand shell only plays a minor role, so be it, but that would have to be reconciled with the overall argument that entropy rules in this assembly process.

On a second and less critical note, it is known that the size distribution of particles has an effect on their assembly, although it is somewhat unclear how strong it is. It would be helpful for readers if the

authors could at least add one sentence on how the distribution of the physical samples will lead to differences between experiments and simulations. The article mentions that "small changes in cube shape already alter the self-assembled superstructures", and while this pertains to the "rounding" of the cubes, it makes me wonder how strong the effect of different particle sizes then is.

There will also a distribution in the size and structure of the particle assemblies (the "balls of cubes") experimentally. I assume that the presented images and reconstructions are representative, but I think the authors should mention whether at least the obvious features of the experimentally observed supraparticles are similar. Best would be SAXS data, of course, and while I would not demand additional SAXS measurements the authors should not withhold SAXS data if they have it.

In summary, this contribution contains a wealth of useful data both from simulation and from experiments that are of very high quality and should be published. The discussion does not make full use of the available data in some parts, however, and I suggest a revision to ameliorate this shortcoming.

Reviewer #3 (Remarks to the Author):

This is an interesting study about the behavior of sharp and rounded nanocubes under spherical confinement. The authors managed to synthesize three types of emulsion droplets filled with nanocubes with different physical (and chemical) composition. Theoretical modeling is used to understand how particle roundness affects the overall structure of these superparticles. The effect shown in the paper is very generic, governed by entropy to a large extent, as shown for two similarly shaped nanocube systems with very different chemical composition.

Clearly, the experimental parameter space for creating cubic particles with various edge sharpness is limited, and the introduced simple hard particle model seems to do an excellent job in capturing the underlying physics of the three experimental systems considered. As far as I see it, this is essentially a packing problem (how many hard cubes one can squeeze in a sphere).

The manuscript is well-written, easy to read, and sufficiently detailed. The visualization of the models and the experimental SPs is very elegant and helpful. I recommend publication in Nature Communications. There are only a few minor issues the authors should address:

1. Possible reasons for anti-correlation shown for the $\alpha=0.3$ model are not discussed at all. It seems like a pretty significant effect, which is not present for the experimental system. Is there any other α value among those considered which exhibits a similar orientational correlation profile? Or could this be due to some arrested structure, with the slightly more dense (and orientationally less correlated) structures not being accessible for these shapes using the MC simulation protocol? Or the experimental particles can rotate more easily because of the slight polydispersity, or because of the interdigitating ligands, the rounding effect being larger than assumed from the ligand sizes? Providing orientational correlation values for all 10 modeled cases (even as a supplementary figure) would help clear this up.
2. It would be good to have an interactive view available for the $\alpha=0.3$ case as well (shown in Figure 3d and f)
3. Figure 6: the authors should specify that these are cut-through views.

4. Supplementary Methods, page 3: "... in the experiments we observe a simple cubic (SC) core surrounded by several shells" - it is not specified that this is for the sharp cubes.

Typos:

- Supplementary Figure 9 label: correct 'left' to 'right'
- Supplementary Section 2.2: correct 'isoproponal'

1 Reviewer #1 (Remarks to the Author):

In this paper, the authors presented a comprehensive investigation of the self-assembly of cube like particles with spherical confinement using both experiments and computer simulations. This problem is of relevance to self-organization of nanoparticles in drying emulsion droplets, and the resulting assembly structures determine their physical properties. Motivated by experimentally realizable particles, the authors investigated both perfect cubes as well as cubes with rounded corners, which are parameterized by the asphericity α . As the particle shape varies, the authors observed a rich spectrum of assembled superstructures. For example, at the cube limit, strong face-face orientational correlations are observed and a significant portion of the superstructure possess “cubic” symmetry as observed in bulk assembly, but with topologically necessary defects. On the other hand, for rounded cubes, the superstructure typically exhibits an icosahedral symmetry, with even richer local correlations. The paper is of great topical interest and well written. It demonstrates the richness of the assembled superstructures via the interplay of particle shape and geometrical confinement, and provides guidance for experimental realization of various superstructures. Therefore, its publication is recommended. However, the authors may want to consider the following suggestions and comments, which are all optional and up to the authors.

We would like to thank the reviewer for the appreciation of our effort on the current work.

Apparently, the superstructures generated using MC simulations were associated with a very slow shrinking rate for the spherical confinement. How does this shrinking rate affect the overall symmetry and local order of the packing? It can be expected that less ordered packing will be obtained for rounded cubes, but does the overall structure still possess the icosahedral symmetry, with a “glassy” sub-regions?

We thank the reviewer for the comments. We think that an extensive study of the glassy behaviour and the effect on the compression rate would

be very interesting but is beyond the scope of this manuscript. Furthermore the MC simulations used here are not ideal to investigate dynamical phenomena such as the glass transition. However, we have performed several simulations at higher compression rate, although we have not studied the effect on the compression rate systematically. When the rate is 10-100 times faster the particles still self-assemble into a crystalline lattice and in icosahedral assemblies, however, the number of defects increases. When the rate is 10,000 times higher the crystalline and the icosahedral symmetry are both lost (for rounded cubes), but we have not performed an extensive study and it might be that an intermediate regime exists. We have added a short discussion as well as snapshots of less ordered packings obtained at high compression rate to the Supplementary Information:

The superstructures lose their order when the compression rate in the simulations is much higher ($\sim 10,000$ times) than the rates used to obtain equilibrated configurations. Supplementary Fig. 11 shows two typical configurations that were obtained after a fast compression.

Did the authors investigate the effects of varying the number of particles on the overall assembly structures? In the infinite system limit, it can be imagined that the effect of the spherical confinement will be only localization to a very thin surface layer of the entire system. And within the system, the structure should be very similar, if not identical, to the bulk assembly structure. For finite system, the effects of the confinement would penetrate deeper in the system, and it would be very interesting to quantify this “penetration depth” if possible. For example, in Fig.6 the authors showed the fraction of local structures consistent with the bulk assembly. It would be also interesting to show such a fraction, but as a function of distance to the sphere confinement center, if the authors also think this would be helpful.

We thank the reviewer for the suggestion on the “penetration depth” quantification. We have varied the number of particles inside the SP for the case of sharp cubes ($\alpha = 1.0$) as can be seen in Supplementary Fig. 10. We agree with the reviewer that the penetration depth might be a more insightful parameter than the fraction of particles that is aligned and plotted the penetration depth in the Supplementary Information. We have added a discussion along the lines of a penetration depth and would like to thank the reviewer for this helpful comment. We can easily obtain the penetration

depth p from the fraction of aligned particles f (see Supplementary Section 3),

$$p = R \left(1 - (1 - f)^{\frac{1}{3}} \right),$$

where R is the radius of the SP. We think that the trend of p and f is intriguing and warrants further investigation. As one could naively expect that f would decrease and p would approach a finite bulk value for larger SPs. However, we think that this more detailed investigation would be more suitable for a separate publication. We have added an additional plot of the penetration depth p to the SI and included the following discussion:

We can convert the fraction f to a length scale to obtain the thickness of the layer effected by the confining surface $p = R \left(1 - (1 - f)^{\frac{1}{3}} \right)$, where R is the radius of the SP. This length scale p seems to grow almost linearly with the size of the SP. This counterintuitive result is probably caused by the decrease in the curvature of the interface as the SP size increases. This results in a decrease in the free energy cost of bending crystalline layers parallel to the interface.

For spheres a more detailed analysis can be found in reference 14. We expect that our rounded cubes will behave in a similar fashion. In this case there is a sharp transition around 40,000 particles when the structure transitions from an icosahedral packing (in which the surface induced defects reach the core) to a structure in which the center of the SP is a bulk like crystal and only a thin surface layer is effected by the curved surface. It might be possible to extract this penetration depth from these larger systems, however, we do only have simulations up to 10,000 particles, in which an icosahedral structure is formed. We think that also in this case an analysis in terms of penetration depth could indeed be very insightful and we might add this to a future publication in which we can also provide experimental measurements of this quantity in SP of different sizes.

It would be helpful to put forth concise qualitative organizing principles that summarize the findings.

We thank the reviewer for the suggestion. We have added the following concise summary to the conclusion of the manuscript:

There are two organizing principles that we expect to hold more generally for hard convex shaped particles. The first principle is that flat faces tend to align particles, both with respect to each other as well as with other

flat surfaces. This alignment is strong as it limits 2 out of the 3 rotational degrees of freedom and plays an important role at high volume fractions. The second principle is that sharp corners play an important role in keeping neighbouring particles at a distance, and even a small amount of rounding of these corners can drastically change the phase behaviour. Furthermore, in confinement there is a competition between the bulk and the surface induced structures, both orientationally as well as positionally. Due to this competition equilibrium defects emerge that can have a complex topology whose details will depend on the symmetries of the equilibrium crystal structure and the free energy cost of the different defect types.

When the reviewer quickly looked at the figures for the first time, it seemed that the authors investigated dense packing of the particles on Riemann (spherical) surface (instead of a bulk packing with spherical confinement). How does the outer layer of the packing in the confinement differ from that on the spherical surface? Any observed difference should stem from the effects of the particles inside in the confinement, but it would be interesting to comment on this point as well.

We would like to thank the reviewer for this suggestion and have added the following to our discussion:

Li *et al.* have studied the 2D assembly of squares on a spherical surface⁵⁴. The defect positions observed by them differ from the positions we observe for sharp cubes. For sharp cubic particles we observe 8 defects in a cubic arrangement (positioned at the vertices of a cube) while they observe defects in an square antiprism arrangement (positioned at the vertices of a square antiprism or a twisted cube). In 2D SA on a curved surface the position of the defects is solely determined by the interactions between the defects through this 2D layer. As shown by Li *et al.* in this case the repulsion between the defects will cause the defects to position themselves as far away from each other as possible and form a square antiprism arrangement. In our case the defects do not only interact through the surface but also through the particles layers below the surface layers which includes the cubic lattice in the bulk of the SP. It is the interaction with this cubic lattice that favors the positioning of the topological defects near the vertices of this cubic lattice and causes the cubic arrangement of defects. This is strengthened by the fact that we always observe the surface defects near the corners of the central cubic lattice. Furthermore this cubic arrangement of the defects transitions to a

square antiprism arrangement of defects when the SP loses its SC core when the cubes become more rounded.

2 Reviewer #2 (Remarks to the Author):

The manuscript by Wang et al. entitled “The interplay between spherical confinement and particle shape on the self-assembly of sharp and rounded cubes” discusses the assembly of cubic inorganic nanoparticles inside evaporating emulsion droplets. The authors, well known experts in the field, combine simulation and experiments and provide a well-written and clearly illustrated account of their work. This is an important contribution in my opinion, in particular since there is an ongoing discussion in the field on the relative magnitudes of entropic versus enthalpic contributions in nanoparticle assembly, and this contribution gives important clues towards a complete picture. There are some shortcomings, mainly when it comes to the comparison of experiment and simulation and possible implications for the understanding of the assembly process. But the quality of data is such that I believe that the manuscript should be accepted after suitable revision.

We would like to thank the reviewer for acknowledging the importance of our work.

The introduction mentions several important publications on the assembly of particles in confined spaces but misses some that I find relevant in this context. I think the authors should mention the recent work of Agthe published in a Nano Letter in 2016 that followed the assembly of maghemite nanocubes in a levitated droplet in situ, and possibly discuss their results in view of the experimental (time-dependent) scattering data. It could also be useful to mention that LaCava et al argue, in a Nano Letter from 2012, that their nanoparticles assume minimum energy arrangements that are similar to LennardJones clusters (and thus not entropically dominated).

We thank the reviewer for pointing out the references. We have added the work by Agthe *et al.* as a reference (Ref. 48 in the revised main text). Also, we cited Lacava’s work (Ref. 20 in the revised main text):

Lacava *et al.* attributed the formation of icosahedral clusters of gold NPs to energetic interactions²⁰, similarly as was *e.g.* found for atomic systems interacting through a Lennard-Jones potential.

The technical details both in the main article and in the supporting information are very thorough and clear, both in terms of experimental procedures and simulation methods; the authors should be lauded for their diligence. I found the discussion of the ligand shell structure on page 3 of the SI particularly interesting, although it leads to some additional questions (see below). If the authors have TGA data on the particles or other data that could indicate the ligand shell density and the amount of free surfactant and other molecular solutes (if any) it would be useful to add it.

We thank the reviewer for the suggestion. Unfortunately we do not have TGA data available for this system but the ligands are discussed in more detail now in the revised manuscript. We agree with the reviewer that this system would be suitable for a more detailed investigation on the role of the ligands, a topic receiving a lot of attention recently in the nanoparticle science community. However, the main subject of this manuscript is not the role of the ligands and therefore we think that TGA data is not required, and expect based on literature data that also in our case the grafting density which was found to be fairly constant as a function of particle shape to be close to an earlier determined value ~ 3 oleate chains/nm² [Dehsari *et al.* “Combined experimental and theoretical investigation of heating rate on growth of iron oxide nanoparticles.” Chem. Mater. 2017, 29, 9648-9656].

We also added following to the Methods Section:

All NCs used in our current work are washed well before SA according to their original experimental protocols. Thus one can assume the amount of free ligands is negligible after proper washing so that they do not play a significant role during the SA.

In the main part, the authors clearly show that rounded corners have a strong effect on particle assembly. Their discussion of this point is convincing and combines simulations and experiments in a very useful way. It would possibly have been more didactic to structure the two central figures 2 and 3 more similar. I was first confused because panels c and d look similar but are once outer/inner views and once experiment/simulation but this is probably a matter of taste. I do find it very important, however, to clearly indicate whether the data in figure 5 is based on simulation and experiments and ideally to compare both. Both datasets should be available, why not plot them in the same way? More in general, the discussion should say more on the similarities and differences between simulations and experiments.

We thank the reviewer for the suggestion. We have more clearly labeled the simulation and the experimental data in the figures and now plot them such that it should be easier to compare the simulation and the experimental datasets. We also explained differences between simulations and experiments:

The very weak anti-alignment observed in the computer simulations for $\alpha < 0.8$ is absent in the experiments. We expect that this is caused by the slight deformation (flattening of the $\{111\}$ facets) of the experimental SP.

The authors correctly write (in the SI) that the sensitivity of assembly on particle shape (and perhaps interaction?) could be used to understand the assembly process better. A more stringent comparison in this manuscript could give first indications on the role of the ligand shell, for example, even if detailed investigations are to follow in a forthcoming contribution.

It is certainly possible that the ligand mediated interaction is not perfectly hard, but either slightly soft repulsive or attractive. However, it has been shown (for spheres) that if the softness is only short ranged (roughly less than one tenth of the total diameter, as will be the case here) the phase behaviour and dynamics can be mapped perfectly on the behaviour of hard particles [Filion, Laura, *et al.* "Simulation of nucleation in almost hard-sphere colloids: The discrepancy between experiment and simulation persists." The

Journal of Chemical Physics 134.13 (2011): 134901.]. The same holds for weakly attractive particles, if the attraction is only short ranged it can be mapped [Noro, Massimo G. and Daan Frenkel. “Extended corresponding-states behavior for particles with variable range attractions.” The Journal of Chemical Physics 113.8 (2000): 2941-2944.] onto the well known phase diagram of adhesive hard particles [Tejero, Carlos F. and Marc Baus. “Freezing of adhesive hard spheres.” Physical Review E 48.5 (1993): 3793.], which is nearly identical to the hard sphere case as long as the attraction is weak. We expect that the same holds for non-spherical particles. Thus there exists a regime in which the interaction potential can be varied without effecting the phase behaviour significantly. The good correspondence between our simulations and experiments tells us that we are inside this regime. The sensitivity of the phase behaviour to small variations in rounding offers a route to obtain more information on ligand mediated interactions and the effective rounding of the particles (as mentioned in the Supplementary Information), but that is beyond the scope of the current manuscript and would require experiments on NP with a wide range of alpha values.

The discussion of the ligand shell is, in my opinion, the only true shortcoming of the manuscript at hand. The authors mention the shell on these particles in multiple instances and correctly point out that they somewhat smoothen the face. They only discuss the ligand shell thickness in the SI, however, and it is not clear which contributions they expect, how “round” the particles with the ligand shell really are, or at least how thick the ligand shell is (the SEMs seem to imply thin shells, but this is the dry collapsed state). Given the discussions on entropic versus enthalpic contributions in assembly, I think that a more thorough discussion would be in order. If the outcome is that the ligand shell only plays a minor role, so be it, but that would have to be reconciled with the overall argument that entropy rules in this assembly process.

We thank the reviewer for the suggestion. We have extended our description of the system and the ligands in the main text as well as the discussion of the effect of the ligands and moved it from the SI to the main text. The new discussion reads now:

The nanocubes are stabilized by ligands, oleic acid molecules in our case. To take into account the effect on the ligands we assume that they form a

shell of constant thickness L around the hard core of the particle. We have obtained the thickness of the ligand layer $L = 1.4$ nm from the particle separation by comparing the simulated radial distribution function $g(r)$ as a function of the particle separation r with the experimentally obtained $g(r)$ (see Supplementary Section 2.3). Taking into account the ligands, the total shape parameter α of the Fe_3O_4 nanocubes and $\text{Fe}_x\text{O}/\text{CoFe}_2\text{O}_4$ nanocubes is 0.68 and 0.30, respectively (see Supplementary Section 2.3 for more details). It is likely that the NP interactions are not exclusively hard and that non-spherical NPs at the droplet interface induce more complex interactions^{12,42}. However, the good correspondence between our experiments and the simulations, in which these interactions are not taken into account, indicates that the experimental SA is largely determined by entropy.

On a second and less critical note, it is known that the size distribution of particles has an effect on their assembly, although it is somewhat unclear how strong it is. It would be helpful for readers if the authors could at least add one sentence on how the distribution of the physical samples will lead to differences between experiments and simulations. The article mentions that “small changes in cube shape already alter the self-assembled superstructures”, and while this pertains to the “rounding” of the cubes, it makes me wonder how strong the effect of different particle sizes then is.

We thank the reviewer for the comments. We have not systematically studied the effect of the particle size distribution. However, we have performed experiments on cubes which have a higher polydispersity. Although we do not have full 3D coordinates, the projected images show the same trends as with the monodisperse particles with only slightly more disorder than the more monodisperse iron oxide cubes. We expect that the situation will be very similar as with hard spheres where a moderate polydispersity only shifts the coexistence densities slightly and does not alter the phase behaviour drastically. So surprisingly, a variation in size is found to be less important than a variation in shape. We expect that this is a result of the sensitivity of the excluded volume to the shape of the particles. A small amount of rounding seems to have a large effect on the excluded volume (the volume swept out by the corners is large) while a change in size has a moderate effect on the excluded volume. But at the moment this is mere speculation and a more detailed discussion, possibly combined with theory

and calculations on the excluded volume, will have to wait till a follow-up study.

There will also a distribution in the size and structure of the particle assemblies (the “balls of cubes”) experimentally. I assume that the presented images and reconstructions are representative, but I think the authors should mention whether at least the obvious features of the experimentally observed supraparticles are similar. Best would be SAXS data, of course, and while I would not demand additional SAXS measurements the authors should not withhold SAXS data if they have it.

We thank the reviewer for the comments. The ones shown are indeed representative, but to make this more clear we have added a sentence in the Method section to indicate the size range of obtained particles.

The size of the self-assembled SPs ranges from ~100 to ~800 nm.

Note that we are interested in the SA within a range of SP sizes, and we have not aimed to obtain a monodisperse distribution. However, microfluidics can be used to obtain monodisperse distributions of SPs (see Ref. 17 in the main text). In addition we have studied the effect on the SP size in computer simulations, see Supplementary Section 3 and Supplementary Fig. 9.

In summary, this contribution contains a wealth of useful data both from simulation and from experiments that are of very high quality and should be published. The discussion does not make full use of the available data in some parts, however, and I suggest a revision to ameliorate this shortcoming.

We would like to thank the reviewer for her/his detailed comments and think that the additional discussion, based on her/his comments, has significantly improved the manuscript.

3 Reviewer #3 (Remarks to the Author):

This is an interesting study about the behavior of sharp and rounded nanocubes under spherical confinement. The authors managed to synthesize three types of emulsion droplets filled with nanocubes with different physical (and chemical) composition. Theoretical modeling is used to understand how particle roundness affects the overall structure of these superparticles. The effect shown in the paper is very generic, governed by entropy to a large extent, as shown for two similarly shaped nanocube systems with very different chemical composition.

Clearly, the experimental parameter space for creating cubic particles with various edge sharpness is limited, and the introduced simple hard particle model seems to do an excellent job in capturing the underlying physics of the three experimental systems considered. As far as I see it, this is essentially a packing problem (how many hard cubes one can squeeze in a sphere).

The manuscript is well written, easy to read, and sufficiently detailed. The visualization of the models and the experimental SPs is very elegant and helpful. I recommend publication in Nature Communications. There are only a few minor issues the authors should address:

1. Possible reasons for anticorrelation shown for the $\alpha = 0.3$ model are not discussed at all. It seems like a pretty significant effect, which is not present for the experimental system. Is there any other α value among those considered which exhibits a similar orientational correlation profile? Or could this be due to some arrested structure, with the slightly more dense (and orientationally less correlated) structures not being accessible for these shapes using the MC simulation protocol? Or the experimental particles can rotate more easily because of the slight polydispersity, or because of the interdigitating ligands, the rounding effect being larger than assumed from the ligand sizes? Providing orientational correlation values for all 10 modeled cases (even as a supplementary figure) would help clear this up.

We appreciate the reviewer for acknowledging our efforts on the current work. We have added the following text to explain the absence of the anti correlation in the experimental data sets. Also, we provided the orientational correlation values for all 10 modeled cases as suggested by the reviewer as a

new figure (Fig. 5) in the main text.

The very weak anti-alignment observed in the computer simulations for $\alpha < 0.8$ is absent in the experiments. We expect that this is caused by the slight deformation (flattening of the $\{111\}$ facets) of the experimental SP.

2. It would be good to have an interactive view available for the $\alpha = 0.3$ case as well (shown in Figure 3d and f).

We thank the reviewer for the suggestion. We have added this to the Supplementary Information (see Supplementary HTML file 4).

3. Figure 6: the authors should specify that these are cut-through views.

We thank the reviewer of the suggestion. We have modified the figure caption. The revised figure description reads:

Cut-through views of a-f) typical configurations of SPs of cubes with different asphericity $\alpha = 0.2, 0.4, 0.5, 0.6, 0.8$ and 1.0 as obtained from simulations.

4. Supplementary Methods, page 3: "... in the experiments we observe a simple cubic (SC) core surrounded by several shells" it is not specified that this is for the sharp cubes.

We thank the reviewer for the comment. We have modified this part. The revised sentence reads now:

However, in the experiment of the sharp Fe_3O_4 nanocubes we observe a simple cubic (SC) core surrounded by several shells, which we also see on the simulations but only for $\alpha > 0.75$ (see Fig. 2 in the main text).

Typos:

- Supplementary Figure 9 label: correct 'left' to 'right'
- Supplementary Section 2.2: correct 'isoproponal'

We thank the reviewer for the comments. We have corrected the typos.

REVIEWERS' COMMENTS:

Reviewer #1 (Remarks to the Author):

The authors have made significant efforts in constructively addressing comments from both reviewers. I recommend its publication in the current form.

Reviewer #2 (Remarks to the Author):

The authors, Wang et al, have submitted a revised manuscript "The interplay between spherical confinement and particle shape on the self-assembly of sharp and rounded cubes" and included detailed replies to the three reviewers. I found both the replies and the revisions very convincing, and I do not think it is necessary to add TGA data at this stage.

Both the role of the "softness" of the particles and the stronger effect of shape when compared to their size that are mentioned in the replies are intriguing, and I would like to know more, but I agree with the authors that they should be topics of future publications.

In summary, I recommend publication of this manuscript without further changes.

Reviewer #3 (Remarks to the Author):

I am happy with the changes made to the manuscript. All my observations have been answered, and, as far as I can tell, the authors also addressed those of the other referees properly. I recommend publication of the article in its current form.